# Behavioral differences between infants at and not at elevated risk for autism during a contingency paradigm

Marcelo R. Rosales[1]*, José Carlos Pulido[2], John Sideris[3], Grace T. Baranek[3], Nina S. Bradley[4], Maja Matarić[5], Beth A. Smith[4,6,7,8]

**1** College of Health and Rehabilitation Sciences, The Ohio State University, Columbus, Ohio, United States of America, **2** Planning and Learning Group, Departamento de Informática, Universidad Carlos III de Madrid, Madrid, Spain, **3** Mrs. T.H. Chan Division of Occupational Science and Occupational Therapy, University of Southern California, Los Angeles, California, United States of America, **4** Division of Biokinesiology and Physical Therapy, University of Southern California, Los Angeles, California, United States of America, **5** Thomas Lord Department of Computer Science, University of Southern California, Los Angeles, California, United States of America, **6** Division of Developmental-Behavioral Pediatrics, Children's Hospital Los Angeles, Los Angeles, California, United States of America, **7** Department of Pediatrics, Keck School of Medicine, University of Southern California, Los Angeles, California, United States of America, **8** Developmental Neuroscience and Neurogenetics Program, The Saban Research Institute, Children's Hospital Los Angeles, Los Angeles, California, United States of America

* marcelo.rosales@osumc.edu

## Abstract

### Introduction

Motor impairments have been reported in infants at elevated likelihood of autism and those later diagnosed with autism. However, empirical studies comparing higher to lower likelihood infants are lacking, limiting our understanding of these motor impairments. This study aimed to determine and describe the behavioral differences between infants at higher (HLA) versus lower likelihood (LLA) of autism during a contingency learning paradigm.

### Methods

Thirty full-term infants (6–9 months of age) at HLA and LLA (n = 15 per group) participated in a contingency learning paradigm. Movements of the infant's right leg activated an infant-sized humanoid robot to reinforce production of right leg movements. Gaze behaviors, the number of times the infants activated the robot, and evidence supporting learning were examined and compared between groups.

### Results

Gaze and motor behavioral data suggest no discernable group differences in terms of overall looking duration, anticipatory gazes, and number of robot activations.

**Data availability statement:** Data cannot be shared publicly because of the University of Southern California Institutional Review Board did not approve data sharing for this study (USC HS-14-00911). Data maybe available from the University of Southern California Institutional Review Board (contact via Fax: 323-224-8389, or email: at irb@usc.edu) for researchers who meet the criteria for access to confidential data.

**Funding:** National Science Foundation [NSF CBET 1706964] (PI: B.A. Smith, Co-PI: M.J. Matarić) Children's Hospital Los Angeles Best Starts to Life Research Support Grant (PI: M.R. Rosales; Mentor: B.A. Smith) The funders had no role in study design, data collection and analysis, decision to publish, or preparation of the manuscript. There was no additional external funding received for this study.

**Competing interests:** The authors have declared that no competing interests exist.

However, four of the infants at HLA displayed visual motor patterns that were qualitatively different.

## Conclusion

Results from our study suggest that infants at HLA and LLA are using similar behaviors to learn a contingency learning task, but heterogeneity within the HLA group is noted and requires further study.

---

## Introduction

Infants at higher likelihood of autism, such as those who have older autistic siblings, have been described in the literature as having delayed motor development [1], atypical motor control [2], and atypical visual behavior [3,4]. Further, it has been shown that children with autism have mild to moderate motor impairments [5] and that these difficulties begin in infancy [1]. However, most of the research that examines motor delays in infants at higher likelihood for autism only describes their motor milestone achievement or overall scores on motor assessments [1,2]. Although those tools are valuable for getting a general sense of motor delays, they do not provide information on why these differences exist, what variability exists within a higher likelihood group, and how practitioners and caregivers can potentially intervene to help improve motor development.

Literature focusing on infant motor control has shown that infants at higher likelihood for autism have balance impairments in a seated position [6] and reaching impairments [2] once those skills emerge. Additionally, in a complex motor reaching task, Ekberg et al. (2016) [7] found that infants at higher likelihood for autism were slower to initiate a reach for a rolling ball compared to infants not at increased risk for autism. Infants from that group also began to reach too late to make contact with the ball, while infants at a lower likelihood timed reaches sufficiently to grab the ball. Together, these papers suggest that infants at higher likelihood for autism have motor impairments in controlling movements that require prospective control and anticipating events in the environment.

Overall, the literature suggests that infants at higher likelihood for autism have difficulties acquiring motor milestones at the widely recognized average ages and difficulty performing these motor skills once they are learned [1]. However, it is unclear if infants at higher likelihood for autism have difficulties learning motor skills or if they utilize different behavioral patterns to learn motor skills (i.e., alternative behavioral strategies for learning). Difficulties in motor learning can lead to later attainment of motor skills or the development of alternative means to perform motor tasks. For example, children who are diagnosed with unilateral upper extremity disabilities due to hemiparetic cerebral palsy or perinatal arterial stroke are at risk of developing developmental disregard. Developmental disregard is the term used to describe when a child with upper extremity disabilities disregards the use of the affected side (i.e., the side with impaired motor abilities) and in turn performs most tasks with

the unaffected side [8]. While the motor impairments for those with upper extremity disabilities are more generally more severe compared to autism, the sensory impairments reported in children with autism could be contributing alternative means to learn motor skills.

An assessment of motor learning in infants at higher likelihood for autism will aid in understanding the heterogeneity that may exist between and within populations. Additionally, a better understanding of motor learning in autism will facilitate the development of more effective early interventions that target specific areas of motor skill acquisition. Last, an evaluation of motor skills can lead to future hypotheses that aim to explain the rationales for difficulties in motor milestone achievement.

In the current study, we used a contingency learning paradigm to compare visual motor behaviors between infants at higher likelihood for autism (HLA) versus those without a familial history of autism (LLA = lower likelihood of autism from a general community sample). We used head mounted eye-tracking, wearable sensors, and video data to evaluate behaviors and tested for differences between these two groups. The current literature supports that infants at HLA have difficulties with motor development and performance. We hypothesized that infants at HLA would have lower performance on the motor paradigm. Specifically, our predictions for between group comparisons were as followed: Infants at HLA will 1) be less likely to learn the contingency paradigm; 2) exhibit fewer instances of anticipatory gaze in expectation of the reinforcement reward; 3) produce fewer reinforcements; 4) have a lower looking duration on the reinforcement agent; and 5) exhibit longer intertrial durations between reinforcements. Additionally, we performed an exploratory analysis to test if the infants at HLA exhibited a similar pattern of visual motor behavior compared to the infants at LLA.

## Methods

### Participants

Thirty-three infants at higher (HLA) (n = 16) and lower likelihood (LLA) (n = 17) of autism were enrolled in the study. They were recruited by fliers, online postings, and word of mouth in the greater Los Angeles area between October 2021 and May 2023. The inclusion criteria were that infants were born full term (>37 weeks gestational age) and were between the age of 6 and 9 months old when recruited. Infants with a first (i.e., sibling or parent) or second degree (i.e., aunt, uncle, grandparent, or step sibling) relative diagnosed with autism were included in the HLA group. Infants without a relative diagnosed with autism were considered at LLA. Last, infants were excluded from the study if they scored lower than the fifth percentile on the Bayley Scales of Infant and Toddler Development (fourth edition) for the average of all three domains tested: Cognitive, Language, and Motor [9], cried continuously for 1 minute of the contingency paradigm, or they had a known visual, hearing, or orthopedic impairment.

Thirty infants completed the paradigm; of those, 26 infants provided usable eye gaze data (13 infants at HLA, and 13 infants at LLA). Three infants were excluded, and the final sample was 30 infants, 15 at LLA and 15 at HLA. One infant at LLA was dropped from the study due to technical problems with the robot during the data acquisition. Two infants, 1 at HLA and 1 at LLA, cried continuously for a full minute during the contingency paradigm so their data were excluded. Lastly, 4 participants (2 at HLA and 2 at LLA) did not tolerate the eye tracker but were able to participate in the contingency learning paradigm. Participant characteristics are summarized in Table 1.

### Procedures

The research was approved by the Institutional Review Board of the University of Southern California (HS-14–00911). A parent or legal guardian signed an informed consent form before their infant's participation in the study. Data were collected at Children's Hospital Los Angeles. During data collection, we obtained the infant's anthropometric data (thigh length and circumference, shank length and circumference, foot length and width, and weight), assessed their motor, cognitive, and language development using the Bayley Scales of Infant and Toddler

**Table 1. Participant characteristics for all infants (Median (Range)).**

| Variable | Lower likelihood for autism (n = 15) | Higher likelihood for autism (n = 15) | P |
|---|---|---|---|
| Age (days) | 210 (183-267) | 202 (184-233) | 0.22 |
| Sex | 7 male, 8 female | 8 male, 7 female | |
| Weight (kg) | 7.5 (6.6-8.7) | 8.4 (6.6-11.1) | 0.02 |
| Bayley-4 Cognitive, Percentiles | 75 (50-95) | 75 (50-91) | 0.81 |
| Bayley-4 Language, Percentiles | 66 (39-94) | 66 (45-81) | 0.92 |
| Bayley-4 Motor, Percentiles | 81 (14-97) | 86 (19-97) | 0.90 |
| Ethnicity | 6 Hispanic/Latino, 9 Not Hispanic/Latino | 12 Hispanic/Latino, 3 Not Hispanic/Latino | |
| Race | 3 Asian, 3 Black/African American, 2 White, 6 Other/Multi-Racial, 1 Declined to answer | 1 Asian, 0 Black/African American, 5 White, 7 Other/Multi-Racial, 2 Declined to answer | |

*Bayley-4: Bayley Scales of Infant and Toddler Development, fourth edition.

Development (version 4) (Bayley-4), and a parent completed the First Year Inventory, Version 3.1 (FYIv3.1) [10]. The Bayley-4 is a standardized observational assessment that assess the motor, cognitive, and language development of children between the ages of 16 days-42 months that yields standardized scores and percentiles based on normative data [9]. The FYI v3.1 is a 69-item parent report questionnaire about infant behaviors that may indicate an elevated likelihood for later neurodevelopmental conditions such as autism [11]. The questionnaire generates risk scores on seven computed factors. The factors are: 1) communication, imitation, and play; 2) social attention and affective engagement; 3) sensory hyperresponsiveness; 4) sensory hyporesponsiveness; 5) self-regulation in daily routines; 6) sensory interest, repetitions, and seeking behaviors; and 7) motor coordination and milestones.

A modified version of the contingency paradigm described in Fitter et al. (2019) [12] was used in this study and a companion study [13]. Briefly, infants were supported in an infant highchair in front of an infant-sized humanoid socially assistive NAO robot (Aldebaran United Robotics Group). Infant participants engaged in a twelve-minute contingency paradigm in which their right leg movement greater than 3.0 m/s$^2$ activated a robot to clap and make laughing sounds. The paradigm consisted of a 2-minute baseline, followed immediately by a 10-second demonstration of the robot preforming three kicks of the right leg, then an 8-minute contingency condition, and finally a 2-minute extinction period. The robot kicking its right leg three time with no sound was chosen as the demonstration, so that the robot did not startle the participants. The infant's right leg movements did not activate the robot during the baseline, demonstration, and extinction phases. The robot was programed to produce clapping and a laughing sounds when the infant activated it during the contingency phase.

Prior to the start of the contingency paradigm, a head-mounted eye tracker (Positive Science) was secured on the infant's head and an Opal sensor (APDM, Inc.) was placed on each arm and leg of the infant. Then, a 5-point calibration using a light-up globe toy was performed for the eye tracker, at a distance of 1.5 meters. i.e., the distance between the seated infant and the robot. Eye gaze data were recorded at 30 frames per second and wearable sensor data were collected at 128 Hz.

Two additional cameras were placed to record the infant interacting with the robot. One camera was placed behind the robot and recorded the infant in their chair. The second camera was placed on the side of the infant and the robot and recorded the contingency paradigm from a side view. Prior to the start of the contingency period and after the 5-point calibration, the light-up globe toy was turned on and off 3 times and simultaneously shown to the two external cameras and to the eye tracker. This was performed to allow us to synchronize all video data during post-processing for data analysis.

## Data preparation & reduction

After the data collection, data from the eye tacker were calibrated and processed using Yarbus software. During processing we imposed a graphic overlay of a target equal to 4-degree radius on the video. This was chosen based on past literature that used a 4-degree radius for the eye tracker [14,15]. In addition, we estimated the accuracy of eye gaze traces for each video. We took 5 frames from each calibration point per video (one for each target) and applied an open sourced accuracy calculator [16] to calculate the distance between the center of the graphic overlay and the target being looked at in the video. The eye gaze data had an average accuracy of 1.3 degrees (SD = 1.0). Therefore, we determined that using a graphic overlay of a 4-degree radius was sufficient for determining where the infant was looking for our eye gaze data.

To inspect the accuracy of eye-gaze calibration and quantify data loss due to blinking or closed eyes, we analyzed eye-tracking data using a custom MATLAB script. For each participant, the total number of frames with a pupil diameter of 0 (indicating blinks or closed eyes) was divided by the total number of recorded frames. This provided the proportion of lost data. For LLA on average, 9.2% of the data were lost due to blinking or participants shutting their eyes (mean = 0.092, SD = 0.098). For HLA on average, 3.9% of the data were lost due to blinking or participants shutting their eyes (mean = 0.039, SD = 0.037).

Eye gaze videos were synchronized with the side view camera's video using ELAN software (ELAN 5.8, Language Archive) and the common starting point (i.e., the light-up globe toy). Additionally, custom Python software was used to identify each time the robot was activated by the infant's right leg movements. The timing for each robot activation was confirmed in our frame-by-frame analysis discussed next.

Video data were analyzed to identify the infant's behavioral state, gaze behavior, and the infant's gaze onset for each activation of the robot. The infant's behavior state was coded over the entire duration of the contingency paradigm for 5 states: sleeping, drowsy, alert, fussy, and crying [17]. Visual behaviors that were coded included: start and end of each gaze on the robot (i.e., looking anywhere on the robot) and each instance the gaze was predictive versus reactive in response to the activation of the robot. Predictive gaze was defined as a visual fixation (3 more frames of no eye movement) established on the robot's torso, hands, or face (i.e., the part of the robot the infant activated with their movement) [14,18]. To avoid inclusion of gaze not motivated by anticipation, only gaze that began within 12 frames preceding the activation was deemed predictive [19]. Gaze that occurred subsequent to the 12 frames was classified as reactive [18] if the visual fixation was located anywhere on the robot during its activation period (60 frames). If the infant did not display a predict gaze or a reactive gaze during a given robot activation, the gaze behavior in that instance was classified as a non-robot look. See Rosales et al. 2025 for more details about the regions of interest [13].

Three video coders were trained on select data sets and had to achieve a reliably of 80% before analyzing data. After reliability was achieved, one third of the data each coder processed was assessed for reliability. Percent of agreement was 94.3% for behavioral state, 87.2% for types of visual gaze, and 95.7% for time spent looking at the robot.

## Determining learning based on leg movement and calculating intertrial duration

A learning threshold was defined based on infant contingency learning studies [20,21]. In those studies, an association was considered learned if the rewarded behavior was produced 50% more during a contingency period compared to a period when no reinforcement was provided. During the baseline and extinction phases of the paradigm, infant right leg movements that met the criteria for robot activations during the contingency phase were identified as 'potential activations' to identify if the infant learned the association. Thus, infant kicks that exceeded an acceleration of 3.0 m/s$^2$ and occurred more than 2 seconds after the last counted movement (i.e., the duration of a single robot activation) counted as potential activations of the robot. Using MATLAB, the number of times an infant would have activated the robot were counted for the baseline and extinction phases. The product of the number of potential baseline activations and 1.5 equaled the learning threshold. If an infant activated the robot above the learning threshold in a two-minute moving window during the contingency phase, they were categorized as a classically-defined learner of the paradigm.

 

We used the onset time of each robot activation to calculate intertrial duration. Intertrial duration equaled the start time of a robot activation minus the start time of the prior activation. For example, if activation 1 happened at 20 seconds into the contingency and then activation 2 happened at 40 seconds into the contingency, then the intertrial duration would be 20 seconds. The average for each infant was calculated and average intertrial duration for each group is reported.

**Visual gaze variable**

The visual gaze measures were the gaze onset for each activation of the robot, looking duration, and proportion of each type of gaze. To calculate the gaze onset for each activation of the robot, the start of each predictive or reactive gaze was subtracted by the start of the robot activation. Negative values were predictive and positive values were reactive. For example, a predictive gaze that occurred at 15 seconds and a robot activation at 15.1 seconds would have a gaze onset of −0.1 seconds (15–15.1 = −0.1). Looking duration was the sum of the time spent looking at the robot during each part of the paradigm. Proportion for each type of gaze was the number of each type of gaze divided by the total number of robot activations.

**Statistical analysis**

Variables were checked for normality. Most variables were not normally distributed and the sample size was deemed small. Therefore, non-parametric tests and descriptive statistics were used. All computations for frequency and proportion of each type of gaze, classification of learning, and onset of an infant's gaze on the activation of the robot were computed using custom MATLAB programs and exported to be analyzed in SPSS (v.27). The following descriptive statistics were compared using a two-sided Wilcoxon Rank Sum test to test for differences between infants at HLA and LLA: behavioral state, age, weight, and Bayley percentiles for the cognitive, language, and motor scales.

A Chi-squared test was used to compare the proportion of learners in each group. Cohen's W was used to describe the effect size and used to calculate the needed sample size to determine a statistical difference. We used the following values for the Cohen's W to describe the effect size: less than 0.1 is a low effect, greater than 0.3 is a moderate effect, greater than 0.5 is a large effect [22]. Additionally, we used Spearman correlations to determine if being classically defined as a learner or non-learner was associated with the percentiles scores from the Bayley-4.

We used a two-sided Wilcoxon Rank Sum test with Bonferroni adjustment to compare the visual motor variables between the infants at HLA and LLA. Additionally, effect sizes were also calculated using Pearson's r. The explanations for Pearson's r were the following: small effect was $|r| > 0.1$, moderate effect $|r| > 0.3$, and large effect $|r| > 0.5$ [22].

To better examine evidence of autistic traits in our sample, we examined the factor scores from the FYIv3.1. We computed the factor scores for each infant using the methods described by Baranek et al (2022) [11]. We combined our data with an age-matched sample (all aged 6 to 7.99 months) from a subset of the data presented in Baranek et al (2022) [11] and applied their model to combined data; while the model was structured to match Baranek et al. (2022) [11], all model parameters were freely estimated. Factor scores were estimated for each infant in our data set with values being centered at zero and positive values indicated greater likelihood of autistic traits. Higher scores on the factors indicate more autistic features. Using a two-sided Wilcoxon Rank Sum test, potential differences between the two group of infants in our study (i.e., HLA and LLA) were assessed using the computed factor score. We also descriptively compared each group's factor scores with the normative averages computed from extant data using an aged-matched subsample from Baranek et al (2022) [11]. The age-matched subsample (n = 53) were infants from the general population [mean age in months (standard deviation) = 7.28 (0.43)] without evidence of a later diagnosis of autism or other developmental disability by age 3 years.

Last, to descriptively explore the individual data further, we plotted the number of gaze behaviors, types of gaze, and duration spent looking at the robot in one-minute blocks across the contingency phase. This was done as an exploratory analysis to consider how the infants at HLA may have used their visual motor behavior to perform the paradigm.

Specifically, we wanted to see if infant at HLA exhibited different behavioral patterns from infants in LLA. Infants who appeared to show unique patterns of visual motor behavior are discussed further using their FYIv3.1 factor scores and performance variables from the contingency learning paradigm (see Results section "Visual motor patterns of learning in infants at elevated risk").

## Results

### Within-group learning

The proportion of infants who were categorized as classically-defined learners (see Methods) was 0.27 in the HLA group and 0.4 in the LLA group. Table 2 reports the number of infants per group and their learning classifications. A chi-squared test showed no significant difference in the proportion of classically-defined learned between the groups ($x^2 = 0.60$, $p = 0.44$). The effect size for the chi-squared test indicated a low effect (Cohen's $w = 0.14$).

Additionally, using Spearman's correlations, no statically significant associations were found between learning classification and cognitive ($r = 0.27$, $p = 0.14$), language ($r = 0.07$, $p = 0.71$), and motor percentiles ($r = 0.28$, $p = 0.14$) from the Bayley-4, or between learning classification and group membership (i.e., HLA vs. LLA) ($r = 0.14$, $p = 0.46$).

### Amount of predictive gaze, reinforcements, looking duration and intertrial duration

We found no significant differences in the visual motor behavior between the infants at HLA versus LLA. The effect sizes for the majority of behavioral variables indicated that there were no effects except for looking duration during the extinction phase. During the extinction, infants at HLA trended to look longer and the effect size was moderate. The medians, ranges, and effect sizes for all variables are reported in Table 3.

**Table 2. Contingency table depicting the number of classically defined learners and non-learners in each group. LLA = Lower likelihood of autism, HLA = Higher likelihood of autism.**

|  | LLA | HLA | Total |
|---|---|---|---|
| Learners | 6 | 4 | 10 |
| Non-learners | 9 | 11 | 20 |
| Total | 15 | 15 | 30 |

**Table 3. Amount of reinforcements, predictive gazes, duration of looking, and intertrial duration. Median (range) are presented in the table.**

| Variable | LLA (n = 15) | HLA (n = 15) | p-value | Pearson's r |
|---|---|---|---|---|
| Potential Activations during Baseline | 18 (5-42) | 25 (8-31) | p = 0.94 | 0.02 |
| Peak Contingency Block | 32 (13-48) | 34 (13-46) | p = 0.65 | 0.09 |
| Potential Activations during Extinction | 23 (8-47) | 30 (13-44) | p = 0.20 | 0.24 |
| Reinforcements | 77 (31-160) | 81 (32-144) | p = 0.68 | 0.08 |
| Proportion of Predictive Gazes | 0.47 (0.33-0.72) | 0.42 (0.21-0.76) | p = 0.45 | 0.15 |
| Looking-Baseline (s) | 17.67 (2.39-68.10) | 20.29 (6.22-51.85) | p = 0.76 | 0.07 |
| Looking-Contingency (s) | 255.93 (152.88-297) | 267.34 (154.91-355.58) | p = 0.31 | 0.21 |
| Looking-Extinction (s) | 7.89 (0.00-43.27) | 13.22 (0.00-26.66) | p = 0.13 | 0.30 |
| Median Gaze Time | −0.31 (−0.4- 0.75) | −0.24 (−0.4- 0.63) | p = 0.26 | 0.226 |
| Average Intertrial Duration (s) | 5.96 (2.98-4.87) | 5.7 (3.31-14.02) | p = 0.68 | 0.08 |

Note: Eye gaze data was not collected on 2 infants from each group due to compliance with the eye tracker. The sample size for eye gaze variables is 13 for HLA and 13 for LLA.

## Evidence of autistic traits amongst the community and elevated risk for autism groups

The caregivers of 9 infants at LLA and 9 infants at HLA filled out the FYIv3.1. No significant differences were found between infants at HLA and LLA for any of the seven factors of the FYIv3.1 (see Table 4). Additionally, the average factor scores from both group in our data set (i.e., LLA and HLA) were similar to the values for the age-matched normative sub-sample from Baranek et al. (2022) [11].

## Visual motor patterns of learning in infants at elevated risk

We visually inspected the data to examine patterns of visual motor behavior that infants exhibited while engaged in the paradigm. Fig 1 illustrates some of our post-hoc observations for 5 infants (1 HLA, 4LLA). Overall, most of the infants at HLA (9 of the 13 infants at HLA) compared to the infants at LLA exhibited visual motor behaviors similar to those of most infants at LLA (13 of 15). Specifically, there was a 1-minute interval during the contingency phase when each of these 22 infants visually anticipated most of the activations (see predictive block, Fig 1). Subsequent to this 'peak' in anticipation, each infant exhibited greater gaze variability responses (anticipatory, reactive, no gaze).

In contrast, there were 4 infants at HLA who had a different phenotypic presentation – they did not exhibit the predictive block or exhibited a unique behavioral pattern. Infants NL2, NL5, and L2 (NL = classically defined non-learner, L = classically defined learner) were identified as the infants who did not exhibit the predictive block (see Fig 1), and Infant L4 had a unique behavioral pattern. Infants NL2 and NL5 increased their activations of the robot without visually anticipating the robot activations and infant L4 had an attentive strategy, where they visually fixated on the robot and had mostly predictive gaze throughout the paradigm. Infant L2 did not have a recognizable pattern in their behavioral data.

Individual data from these four infants at HLA were explored further against the average data for the infants at LLA. Infants NL2, NL5, and L2 displayed lower durations of looking at the robot during the contingency phase and a greater proportion of non-robot and reactive looks towards their activation of the robot (Table 5). Additionally, L2 activated the robot fewer times compared to all other infants in the sample, while L4 produced the greatest number of predictive gaze behaviors and looked at the robot substantially more during the contingency phase compared to all other infants (Table 5).

Three of these four infants had caregivers who filled out the FYIv3.1. Infant NL2 had factor scores that were similar to the normative age-matched sample. Infant NL5 had higher factor scores indicating more autistic traits for communication, imitation, and play (factor score = 1.18) and motor coordination and milestones (factor score = 1.28). Lastly, infant L4 had higher factor scores (i.e., they were greater than 1) for 5 of seven factors: communication, imitation, and play (factor score = 1.064), social attention and affective engagement (factor score = 1.417), sensory hyporesponsiveness (factor score = 1.472), self-regulation in daily routines (factor score = 1.534), and motor coordination and milestones (factor score = 1.377).

**Table 4.** FYIv3.1 factor score means (standard deviations) from the study sample groups (HLA and LLA) and comparison to norms from an aged-matched subsample from Baranek et al. (2022) [11]. P-values are for the comparison between the LLA and HLA groups from the study.

| FYIv3.1 Factor | LLA sample (n = 9) | HLA sample (n = 9) | Norms from age-matched subsample (n = 53) | P-value |
|---|---|---|---|---|
| Communication, imitation, and play | −0.04 (0.74) | 0.39 (0.73) | −0.08 (1.00) | $p = 0.19$ |
| Social attention and affective engagement | 0.07 (0.72) | 0.20 (0.67) | −0.06 (1.00) | $p = 0.80$ |
| Sensory hyperresponsiveness | −0.67 (1.1) | −0.71 (0.60) | 0.23 (0.91) | $p = 0.80$ |
| Sensory hyporesponsiveness | 0.14 (0.49) | 0.16 (0.68) | −0.06 (1.02) | $p = 0.93$ |
| Regulation in daily routines | −0.21 (0.75) | −0.16 (0.87) | −0.04 (0.89) | $p = 0.34$ |
| Sensory interest, repetitions, and seeking behaviors | −0.52 (0.83) | −0.52 (0.47) | 0.16 (0.94) | $p = 0.73$ |
| Motor coordination and milestones | −0.02 (0.74) | 0.31 (0.88) | −0.06 (0.98) | $p = 0.34$ |

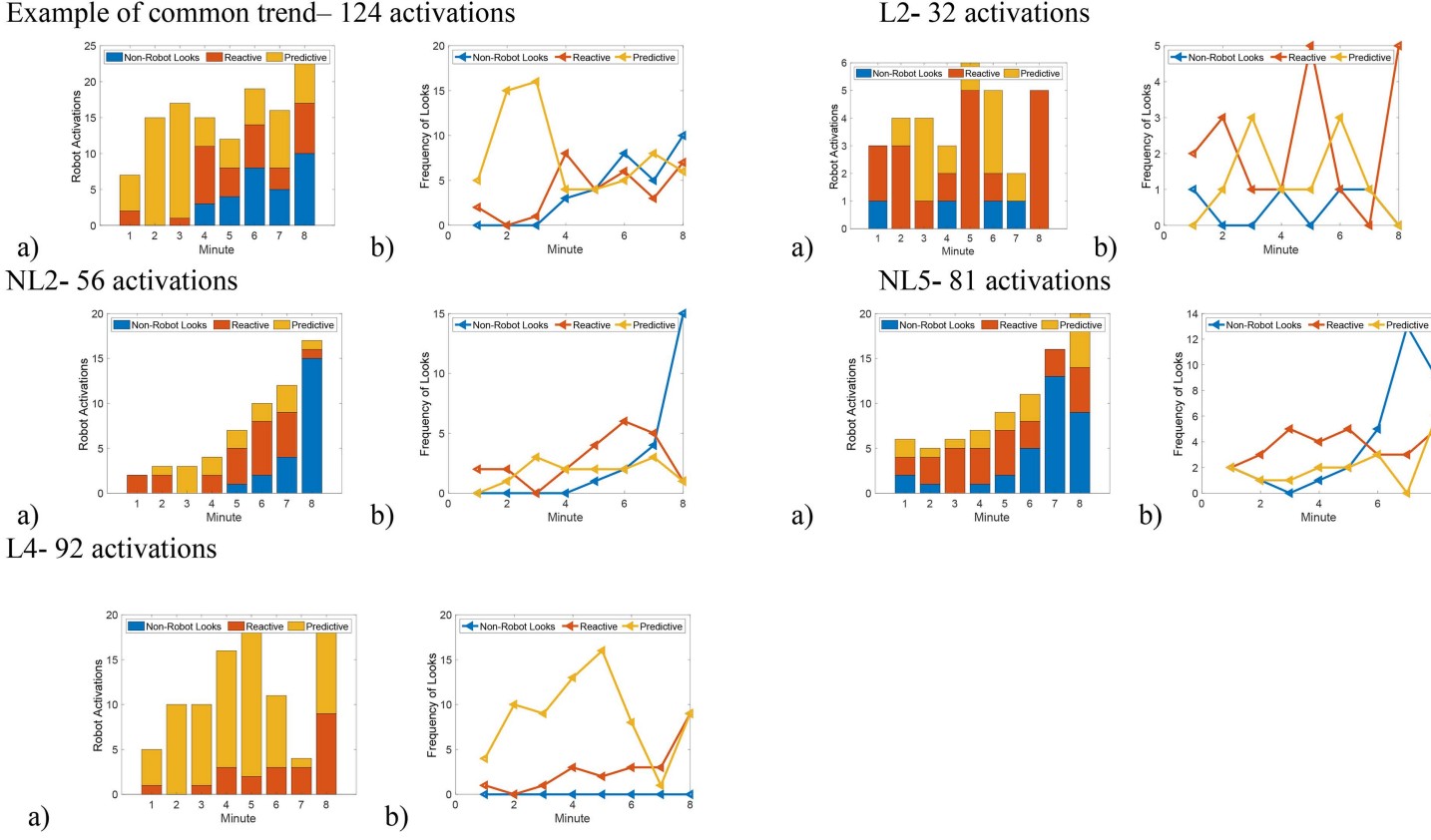

**Fig 1. Individual plot for an example infant and the four unique infants.** The letter ID indicated if they are a classically defined as a learner (L) or a non-learner (NL). Graph a) is a bar graph that plots the total number of activations on the y-axis and the minute blocks of the contingency period on x-axis. Graph b) is a line graph that plots the frequency of the type of gaze in the y axis for the minute blocks described in graph **a**. The colors denote the types of gazes (blue = non-robot looks, red = reactive looks, yellow = predictive looks).

**Table 5. Average (standard deviation) behavioral measures from the contingency data for the infants at LLA and four individuals at HLA for autism.** These four infants (NL2, NL5, L2, and L4) were identified to have different behavioral patterns during the paradigm. The letter ID indicated if they are a classically defined as a learner (L) or a non-learner (NL).

| Variable | Infant Average (SD) (n=15) | NL 2 | NL 5 | L 2 | L4 |
|---|---|---|---|---|---|
| Reason for increased likelihood of ASD | | Infant sibling | Second degree relative | Infant sibling | Infant Sibling |
| Infant Robot Activations | 82 (32) | 56 | 81 | 32 | 92 |
| Proportion of Non-Robot Looks | 0.17 (0.15) | 0.39 | 0.41 | 0.13 | 0 |
| Proportion of Reactive Gazes | 0.35 (0.16) | 0.39 | 0.37 | 0.56 | 0.23 |
| Proportion of Predictive Gazes | 0.48 (0.17) | 0.21 | 0.22 | 0.31 | 0.76 |
| Median Timing of Gaze on the initiated Robot activation (s) | −0.09 (0.37) | 0.49 | 0.54 | 0.32 | −0.35 |
| Looking Baseline (s) | 23.02 (17.11) | 17.77 | 12.69 | 27.44 | 18.8 |
| Looking Contingency (s) | 250.78 (53.09) | 190.61 | 154.91 | 199.07 | 355.58 |
| Looking Extinction (s) | 11.78 (9.90) | 0 | 18.77 | 26.66 | 10.46 |
| Proportion of Time Alert Baseline | 95.12 (12.40) | 100 | 100 | 100 | 100 |
| Proportion of Time Alert Contingency | 97.50 (6.19) | 98.69 | 100 | 100 | 100 |
| Proportion of Time Alert Extinction | 88.26 (23.59) | 62.4 | 100 | 100 | 100 |

## Discussion

Previous research suggests that infants who go on to be diagnosed with autism utilize visual behaviors in atypical ways [3,4] and show evidence of motor delays [1,5,23,24]. Thus, our study hypothesized that there may be differences in motor performance among HLA and LLA infants during a contingency learning paradigm. Contrary to our hypotheses, no significant group differences were detected in this sample of 15 HLA and 15 LLA. The proportion of classically-defined learners and non-learners were similar among the infants at HLA and LLA; and both groups (on average) displayed similar visual motor behaviors while engaged in the paradigm. Therefore, these results suggest that infants at HLA and LLA display similar behaviors while learning a contingency learning paradigm between the ages of 6 and 9 months. However, while group comparisons were not significantly different, exploratory descriptions of visual motor behaviors during the paradigm and individual assessment data noted phenotypic heterogeneity within the HLA group, which raises questions for further study.

The lack of significant group differences could be due to several reasons. One possible explanation is that our sample may have contained very few or no infants who will go on to be diagnosed with autism. When comparing the data from our assessment of prodromal autistic traits at 6–9 months of age, there were no statistical differences at the group level for either the Bayley-4 subset of items, or the FYIv3.1 factor scores, and we noted that the overall scores for the groups were also quite comparable to norms for this age group based on Baranek et al. (2022) [11]. The literature shows that having a sibling with autism increases the infant's likelihood of being later diagnosed with autism to 18% (versus 2–3% for community samples) [25–27]. Thus, out of our sample of 15 HLA infants we would anticipate that only 2 or 3 infants may go on to receive a diagnosis of autism; further longitudinal follow-up is needed to confirm diagnostic outcomes.

Although recent literature suggests that infants who are diagnosed with autism later on in life have mild to moderate motor delays [5], it is also possible that the motor paradigm chosen for this study was not sensitive enough to detect more subtle motor differences. Behavioral descriptions of motor control impairments in infants who go on to be diagnosed with autism have been shown in paradigms that use quantifiable measures that are difficult to assess by visual observation. For example, in Ekberg et al. (2016) [7], researchers found that infants who were later diagnosed with autism, compared to those who were not, had difficulties timing their grasp of a ball that was moving down a ramp. Therefore, an analysis of motor learning behavior during an error based task, for example, may show more distinctions between infants at LLA and ER.

Additionally, another endeavor that could highlight motor learning differences between HLA and LLA infants would be to examine how infants use adaptive behaviors to learn motor skills. Autistic children experience sensory information in variable ways that can result in the avoidance or seeking selective sensory information and experiences [28]. Autistic traits regarding sensory processing potentially lead to different motor adaptations and developmental profiles in preschool and school-age outcomes [29]. A better understanding of how sensory differences impact motor learning adaptations could aid in the development of more efficacious interventions for HLA infants with these phenotypic variations.

In this study, we explored whether infants at elevated risk for autism displayed different phenotypic patterns of visual motor behavior during the paradigm than infants at community risk and then analyzed their assessment scores in depth. This analysis found that 8 of the infants displayed similar patterns of visual motor behavior and 4 infants did not, pointing to heterogeneity in the sample that requires further study. Two of the four infants (NL2 and NL5) were potentially engaged in the paradigm using a non-looking adaptation (i.e., they displayed less looking, greater proportion of non-robot looks, and a greater proportion of reactive looks) and could have sought the sound of the robot instead of looking at the robot. Another infant (L2), could have been displaying an avoidance pattern since this infant did not activate the robot as much as the other infants in the study, or potentially that infant did not move very much and thus did not activate the robot as often. Finally, infant L4 displayed hightened attnetion to the robot and did not look elsewhere while the robot was responding to their kicks, thus potentially displaying "sticky attention" to the robot. It should be noted that these four infants were in the alert behavioral state for the vast majority of the paradigm and showed few occurrences of fussiness and crying while the robot was being activated by their movements. Interestingly, although all 4 of these infants scored within the average range for the Bayley-4 subset of items, two of the 4 infants had elevated scores on the FYIv3.1, an autism specific

screener for this age range. Future studies are needed to determine if the phenotypic heterogeneity seen within the HLA group is predictive of later diagnosis through longitudinal follow-up methods and gold-standard diagnostic instruments.

## Limitations

It is possible that our study lacked power to find statistically significant group differences due to our sample size of only 15 infants in each group. In Table 3, it appears that HLA infants in this sample are trending towards being less likely to be categorized as classically defined learners of the motor paradigm. A retention test or transfer paradigm could be considered in order to confirm that learning occurred; however, retention or transfer paradigms are difficult for autistic individuals with sensory difficulties [28]. Using this assessment of learning might not be appropriate for HLA infants, but our analysis of motor learning adaptations could highlight how infants who go on to be diagnosed with autism use behavioral adaptations to learn motor skills. These concepts should be re-evaluated in future studies when learning is best determined.

## Conclusion

In summary, while our hypotheses of group differences in motor learning between HLA and LLA infants were not supported, our study offers guidance for future work. Purposive sampling of infants with elevated scores on screening measures (rather than infant siblings of autistic children more generally) could be useful in future studies to increase the probability of finding significant differences and measuring heterogeneity in motor performance more fully. Given the variety and neurodiversity of autism symptoms, the use of purposive sampling would allow for a selection of individuals that would benefit from findings that are geared towards informing early interventions for the selected group.

## Acknowledgments

The authors thank all of the infants and their caregivers for their participation in the study. We also thank the staff of the Infant Neuromotor Control Laboratory at Children's Hospital Los Angles.

## Author contributions

**Conceptualization:** Marcelo Ramon Rosales, Nina S. Bradley, Maja Matarić, Beth A. Smith.

**Data curation:** Marcelo Ramon Rosales.

**Formal analysis:** Marcelo Ramon Rosales, John Sideris, Beth A. Smith.

**Funding acquisition:** Marcelo Ramon Rosales, Maja Matarić, Beth A. Smith.

**Investigation:** Marcelo R. Rosales, Beth A. Smith.

**Methodology:** Marcelo R. Rosales, José Carlos Pulido, Nina S. Bradley, Maja Matarić, Beth A. Smith.

**Project administration:** Marcelo R. Rosales, Beth A. Smith.

**Resources:** Maja Matarić, Beth A. Smith.

**Software:** José Carlos Pulido.

**Supervision:** Maja Matarić, Beth A. Smith.

**Validation:** Marcelo R. Rosales, Beth A. Smith.

**Visualization:** Marcelo R. Rosales, Beth A. Smith.

**Writing – original draft:** Marcelo R. Rosales, Beth A. Smith.

**Writing – review & editing:** Marcelo R. Rosales, José Carlos Pulido, John Sideris, Grace T. Baranek, Nina S. Bradley, Maja Matarić, Beth A. Smith.

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
