## [Decision Letter · Decision Letter 0]

14 Sep 2025

Dear Dr. Rosales,

Thank you for submitting your manuscript to PLOS ONE. After careful consideration, we feel that it has merit but does not fully meet PLOS ONE’s publication criteria as it currently stands. Therefore, we invite you to submit a revised version of the manuscript that addresses the points raised during the review process.

https://journals.plos.org/plosone/s/submission-guidelines#loc-laboratory-protocols . Additionally, PLOS ONE offers an option for publishing peer-reviewed Lab Protocol articles, which describe protocols hosted on protocols.io. Read more information on sharing protocols at https://plos.org/protocols?utm_medium=editorial-email&utm_source=authorletters&utm_campaign=protocols .

We look forward to receiving your revised manuscript.

Kind regards,

Gökhan Töret

Academic Editor

PLOS ONE

Journal Requirements:

“National Science Foundation [NSF CBET 1706964] (PI: B.A. Smith, Co-PI: M.J. Matarić)

Children’s Hospital Los Angeles Best Starts to Life Research Support Grant (PI: M.R. Rosales; Mentor: B.A. Smith)”

Please state what role the funders took in the study. If the funders had no role, please state: 'The funders had no role in study design, data collection and analysis, decision to publish, or preparation of the manuscript.'

“National Science Foundation [NSF CBET 1706964] (PI: B.A. Smith, Co-PI: M.J. Matarić)

Children’s Hospital Los Angeles Best Starts to Life Research Support Grant (PI: M.R. Rosales; Mentor: B.A. Smith)”

“The authors thank all of the infants and their caregivers for their participation in the study. We also thank the staff of the Infant Neuromotor Control Laboratory at Children’s Hospital Los Angles. This research study was a part of M.R. Rosales’ PhD dissertation work at the University of Southern California. This research was supported in part by a grant from the National Science Foundation [NSF CBET 1706964] (PI: B.A. Smith, Co-PI: M.J. Matarić) and in part by a Children’s Hospital Los Angeles Best Starts to Life Research Support Grant (PI: M.R. Rosales; Mentor: B.A. Smith).”

“National Science Foundation [NSF CBET 1706964] (PI: B.A. Smith, Co-PI: M.J. Matarić)

Children’s Hospital Los Angeles Best Starts to Life Research Support Grant (PI: M.R. Rosales; Mentor: B.A. Smith)”

6. Thank you for stating the following in your Competing Interests section:

“NO authors have competing interests”

Please complete your Competing Interests on the online submission form to state any Competing Interests. If you have no competing interests, please state 'The authors have declared that no competing interests exist.', as detailed online in our guide for authors at http://journals.plos.org/plosone/s/submit-now

7. We note that your Data Availability Statement is currently as follows: All relevant data are within the manuscript and its Supporting Information files.

8. We note that you have indicated that there are restrictions to data sharing for this study. For studies involving human research participant data or other sensitive data, we encourage authors to share de-identified or anonymized data. However, when data cannot be publicly shared for ethical reasons, we allow authors to make their data sets available upon request. For information on unacceptable data access restrictions, please see http://journals.plos.org/plosone/s/data-availability#loc-unacceptable-data-access-restrictions.

Reviewers' comments:

Reviewer's Responses to Questions

**Comments to the Author**

1. Is the manuscript technically sound, and do the data support the conclusions?

Reviewer #1: Yes

Reviewer #2: Yes

2. Has the statistical analysis been performed appropriately and rigorously?

Reviewer #1: Yes

Reviewer #2: Yes

3. Have the authors made all data underlying the findings in their manuscript fully available?

Reviewer #1: Yes

Reviewer #2: Yes

4. Is the manuscript presented in an intelligible fashion and written in standard English?

Reviewer #1: Yes

Reviewer #2: Yes

Reviewer #1: The manuscript was interesting and good written and discussed.

Introduction was good written and include the aim of the study.

Methods were good designed.

Results were good described.

Discussion was good written.

Reviewer #2: The authors studied 6 to 9 months old infants with high or low risk for getting an ASD diagnosis, based on sibling diagnoses, and how they could differ in gaze and motor behaviour in a contingency learning paradigm. They didn’t find expected differences (such as less anticipatory gaze or less overall looking time toward the agent), and they discuss this later on. I found the manuscript to be well written, with the Methods carefully and thoroughly described. I have some major concerns, but more like a possible add-on to the article. I found the overall article very sound and I encourage the publication of null results when the methods is sound and carefully described. This being said, I think the authors should run and report a power analysis before the manuscript being deemed appropriate for publication (see major concern #2).

Major concerns

1. Participants section: It might be obvious to the authors, but as it is not clearly stated I do dare ask: were participants screened for motor related impairments such as cerebral palsy or epilepsy? The fact that the two groups of participants are comparable on the Bayley-4 Motor Percentile should be enough to assert that they are both comparable in basic motor skills anyways, but just to be sure, please answer my question and maybe add it to the main text if relevant.

2. I think the authors should include a power analysis in the methods section, to help readers assess whether the null results should be attributed to sample size or not. I really don’t mind the exploratory nature of this article and the null results, but in my opinion, it has to be accompanied by a power analysis.

3. Paragraph between pages 13 and 14: “trended to look longer”. The p-value is 0.13 so I don’t think we can label it as a trend (of course this is subjective but it also linked to conventions in the field, and the dominant convention would say that trending would be between 0.05 and 0.1).

4. A lot of null results could be explained by the fact that both groups were similar in FYI scores. Maybe it’s the phenotype that counts, and not so much the risk factor of having a sibling with ASD? The authors did a good job on commenting on that in the discussion section already. But maybe the authors could also consider median splitting their sample on FYI overall scores? And run their analyses again? Just as an exploratory analysis? I think this would be a much better addition to the article compared to the zooming in on some participants that is made in some sections and in figure 1 (see my major concern 8).

5. In a related manner, the hypothesis that sensory atypicalities could lead to differences in motor functioning is interesting. Have the authors considered median splitting their sample on high and low scores on FYI dimensions 3, 4 or 6?

6. Page 14 “In contrast, there were 4 infants, etc.”: I don’t think zooming in on these 4 participants really add any clarity. Maybe the authors could consider dropping these sections entirely? Or enclosing them in an insert?

Minor concerns

1. There seems to be more lost gaze data for the LLA group due to blinking or participants shutting their eyes. Is the difference significant between the two groups? If so, what could be the explanation.

2. Not sure I understand the last paragraph of page 9. If I understand correctly, reaching a reliability of 80% would already mean that coders assessed the same video segments. So why is there a second step in which one third of the videos were assessed for inter raters reliability, like again?

3. Typos:

- Page 9, last paragraph, 1st sentence: “reliably” should be “reliability”.

- Page 18, line 3: “avoidance” instead of “avoidence”. Line 5: “attention” instead of “attnetion”

**Do you want your identity to be public for this peer review?** For information about this choice, including consent withdrawal, please see our Privacy Policy

Reviewer #1: No

Reviewer #2: **Yes:**  Matias Baltazar

---

## [Author Response · Author response to Decision Letter 1]

7 Oct 2025

Response to Editor and Reviewers

To all,

Thank you all for the comments and feedback and for making the work stronger. Per the editors request we would like to state the following:

• There was no additional external funding received for this study.

• The authors have declared that no competing interests exist.

Below is an item-by-item list of all the points addressed in your comments. In addition, all revisions are highlighted in the uploaded manuscript document.

Editor,

Thank you for your comments. Below is a response for each point:

o The manuscript was revised to the specification in the template.

• Please provide additional details regarding participant consent. In the ethics statement in the Methods and online submission information, please ensure that you have specified (1) whether consent was informed and (2) what type you obtained (for instance, written or verbal, and if verbal, how it was documented and witnessed). If your study included minors, state whether you obtained consent from parents or guardians. If the need for consent was waived by the ethics committee, please include this information.

o The methods section was revised to specify that written informed consent was provided by a parent or legal guardian. In the prior submission, we provided the IRB approval and consent from.

• Thank you for stating the following financial disclosure: “National Science Foundation [NSF CBET 1706964] (PI: B.A. Smith, Co-PI: M.J. Matarić) Children’s Hospital Los Angeles Best Starts to Life Research Support Grant (PI: M.R. Rosales; Mentor: B.A. Smith)” Please state what role the funders took in the study. If the funders had no role, please state: 'The funders had no role in study design, data collection and analysis, decision to publish, or preparation of the manuscript.'If this statement is not correct you must amend it as needed. Please include this amended Role of Funder statement in your cover letter; we will change the online submission form on your behalf.

o The funders had no role in study design, data collection and analysis, decision to publish, or preparation of the manuscript.

• Thank you for stating in your Funding Statement:“National Science Foundation [NSF CBET 1706964] (PI: B.A. Smith, Co-PI: M.J. Matarić) Children’s Hospital Los Angeles Best Starts to Life Research Support Grant (PI: M.R. Rosales; Mentor: B.A. Smith)”Please provide an amended statement that declares *all* the funding or sources of support (whether external or internal to your organization) received during this study, as detailed online in our guide for authors at http://journals.plos.org/plosone/s/submit-now. Please also include the statement “There was no additional external funding received for this study.” in your updated Funding Statement. Please include your amended Funding Statement within your cover letter. We will change the online submission form on your behalf.

o There was no additional external funding received for this study.

• Thank you for stating the following in the Acknowledgments Section of your manuscript:“The authors thank all of the infants and their caregivers for their participation in the study. We also thank the staff of the Infant Neuromotor Control Laboratory at Children’s Hospital Los Angles. This research study was a part of M.R. Rosales’ PhD dissertation work at the University of Southern California. This research was supported in part by a grant from the National Science Foundation [NSF CBET 1706964] (PI: B.A. Smith, Co-PI: M.J. Matarić) and in part by a Children’s Hospital Los Angeles Best Starts to Life Research Support Grant (PI: M.R. Rosales; Mentor: B.A. Smith).”We note that you have provided funding information that is currently declared in your Funding Statement. However, funding information should not appear in the Acknowledgments section or other areas of your manuscript. We will only publish funding information present in the Funding Statement section of the online submission form.Please remove any funding-related text from the manuscript and let us know how you would like to update your Funding Statement. Currently, your Funding Statement reads as follows:“National Science Foundation [NSF CBET 1706964] (PI: B.A. Smith, Co-PI: M.J. Matarić). Children’s Hospital Los Angeles Best Starts to Life Research Support Grant (PI: M.R. Rosales; Mentor: B.A. Smith)”Please include your amended statements within your cover letter; we will change the online submission form on your behalf.

o The funding information was removed from the acknowledgment section per publishing guidelines.

• Thank you for stating the following in your Competing Interests section:“NO authors have competing interests”Please complete your Competing Interests on the online submission form to state any Competing Interests. If you have no competing interests, please state 'The authors have declared that no competing interests exist.', as detailed online in our guide for authors at http://journals.plos.org/plosone/s/submit-nowThis information should be included in your cover letter; we will change the online submission form on your behalf.

o The authors have declared that no competing interests exist.

• Data Availability

o We agree with the importance of data sharing. However, under our current IRB agreement we are not allowed to share data. Therefore, data is available upon reasonable request to the corresponding author.

• The Reviewers comments did not require more literature to cite.

• Reference list is correct and has been checked.

Review 1,

Thank you for all your comments!

Reviewer 2,

Thank you for all your comments and feedback! Below is an itemized list for revisions to address your major and minor concerns.

• Major Concerns

o Participants section: It might be obvious to the authors, but as it is not clearly stated I do dare ask: were participants screened for motor related impairments such as cerebral palsy or epilepsy? The fact that the two groups of participants are comparable on the Bayley-4 Motor Percentile should be enough to assert that they are both comparable in basic motor skills anyways, but just to be sure, please answer my question and maybe add it to the main text if relevant.

Thank you for pointing this out. We confirm that participants were screened for motor-related impairments (e.g., cerebral palsy, epilepsy) as part of our inclusion criteria. We have clarified this in the Participants section.

o I think the authors should include a power analysis in the methods section, to help readers assess whether the null results should be attributed to sample size or not. I really don’t mind the exploratory nature of this article and the null results, but in my opinion, it has to be accompanied by a power analysis.

We appreciate the reviewer’s suggestion regarding power analysis. We did not conduct a post-hoc power analysis because, once a study has been completed, statistical power is directly determined by the observed effect sizes and p-values, making post-hoc power redundant. As Dorey 2010 (doi: 10.1007/s11999-010-1435-0) points out in their discussion about power calculation, calculating power after data has been collected does not change the fact that novel and relevant finding are being presented. Given that this is an exploratory research study, the data presented here would be used in our future grant proposals for confirming studies. We report effect sizes using Person’s r, which provide more meaningful information about the magnitude and precision of the observed effects. This approach is consistent with current methodological recommendations.

o Paragraph between pages 13 and 14: “trended to look longer”. The p-value is 0.13 so I don’t think we can label it as a trend (of course this is subjective but it also linked to conventions in the field, and the dominant convention would say that trending would be between 0.05 and 0.1).

We changed the wording to state that there may have had a slight tendency instead of trend. While the p-value does not suggest trend in different fields, the effect size does. We report both in the paper and softened the language.

o A lot of null results could be explained by the fact that both groups were similar in FYI scores. Maybe it’s the phenotype that counts, and not so much the risk factor of having a sibling with ASD? The authors did a good job on commenting on that in the discussion section already. But maybe the authors could also consider median splitting their sample on FYI overall scores? And run their analyses again? Just as an exploratory analysis? I think this would be a much better addition to the article compared to the zooming in on some participants that is made in some sections and in figure 1 (see my major concern 8). In a related manner, the hypothesis that sensory atypicalities could lead to differences in motor functioning is interesting. Have the authors considered median splitting their sample on high and low scores on FYI dimensions 3, 4 or 6? Page 14 “In contrast, there were 4 infants, etc.”: I don’t think zooming in on these 4 participants really add any clarity. Maybe the authors could consider dropping these sections entirely? Or enclosing them in an insert?

All comments for the FYI are addressed in this statement. We agree and have conducted additional exploratory analyses; however, we decided against the application of a median split, given potential consequences for study power and inference (e.g., MacCallum, et al., 2002; McClelland, et al., 2015).

• MacCallum, R. C., Zhang, S., Preacher, K. J., & Rucker, D. D. (2002). On the practice of dichotomization of quantitative variables. Psychological methods, 7(1), 19.

• McClelland, G. H., Lynch Jr, J. G., Irwin, J. R., Spiller, S. A., & Fitzsimons, G. J. (2015). Median splits, Type II errors, and false–positive consumer psychology: Don't fight the power. Journal of Consumer Psychology, 25(4), 679-689.

In regard to the zooming in on the 4 participants, we choice to retain this piece in the work. This snapshot illustrates the visual gaze patterns that were exhibited, that are different than the common trend. Further work should be conducted to explore these patterns more and the individual data highlights the potential visual motor patterns that can occur.

• Minor concerns

• There seems to be more lost gaze data for the LLA group due to blinking or participants shutting their eyes. Is the difference significant between the two groups? If so, what could be the explanation.

o We tested for significant differences and found that the two groups were not significantly different according to Wilcoxon Rank-sum test. This is now included in the methods section.

• Not sure I understand the last paragraph of page 9. If I understand correctly, reaching a reliability of 80% would already mean that coders assessed the same video segments. So why is there a second step in which one third of the videos were assessed for inter raters reliability, like again?

o This is the standard coding practices for this type of analysis (see….). The first 80% is a training phase so that coders reach 80% reliably during training. Then a third of their data is checked to ensure that the coders maintained reliable standards. We expanded the section to explain this further. The section now reads: ”Three video coders were trained on select data sets and had to achieve a reliability of 80% before analyzing data (i.e. training prior to processing). After reliability was achieved, one third of the data each coder processed was assessed for reliability, to ensure that reliability was maintained throughout processing.”

• Typos:- Page 9, last paragraph, 1st sentence: “reliably” should be “reliability”.- Page 18, line 3: “avoidance” instead of “avoidence”. Line 5: “attention” instead of “attnetion”

o All suggested typos are fixed.

---

## [Editor Report · Decision Letter 1]

12 Dec 2025

BEHAVIORAL DIFFERENCES BETWEEN INFANTS AT AND NOT AT ELEVATED RISK FOR AUTISM DURING A CONTINGENCY PARADIGM

PLOS One

Dear Dr. Rosales,

Thank you for submitting your manuscript to PLOS ONE. After careful consideration, we feel that it has merit but does not fully meet PLOS ONE’s publication criteria as it currently stands. Therefore, we invite you to submit a revised version of the manuscript that addresses the points raised during the review process.

We look forward to receiving your revised manuscript.

Kind regards,

Gökhan Töret

Academic Editor

PLOS One

Journal Requirements:

Additional Editor Comments:

Thank you for submitting your manuscript to PLOS ONE. Thank you for the revision. For your information, all reviewer concerns have been addressed. The only remaining editorial issue is that funding information still appears in the Acknowledgments section. Please remove it, as funding details must be listed only in the Funding Statement. Once this correction is made, the manuscript will be ready for acceptance.

---

## [Author Response · Author response to Decision Letter 2]

15 Dec 2025

Dear Editor,

Thank you for your comments. Below is a response for each point:

• Comment: Please include your tables as part of your main manuscript and remove the individual files. Please note that supplementary tables (should remain/ be uploaded) as separate "Supporting Information" files

o Response: All tables are now in the main document and have been removed from the submission portal. “Revised Manuscript with Track Changes12_15_2025” is the revised manuscript upload.

Thank you for all your hard work and please let us know if you need anything else from us.

Sincerely,

Marcelo Rosales

---

## [Editor Report · Decision Letter 2]

26 Dec 2025

BEHAVIORAL DIFFERENCES BETWEEN INFANTS AT AND NOT AT ELEVATED RISK FOR AUTISM DURING A CONTINGENCY PARADIGM

PONE-D-25-06495R2

Dear Dr. Rosales,

We’re pleased to inform you that your manuscript has been judged scientifically suitable for publication and will be formally accepted for publication once it meets all outstanding technical requirements.

Kind regards,

Gökhan Töret

Academic Editor

PLOS One
---

## [Editor Report · Acceptance letter]

PONE-D-25-06495R2

PLOS One

Dear Dr. Rosales,

I'm pleased to inform you that your manuscript has been deemed suitable for publication in PLOS One. Congratulations! Your manuscript is now being handed over to our production team.

Kind regards,

on behalf of

Dr. Gökhan Töret

Academic Editor

PLOS One